# Spanish Loyalty and English Prestige in the Linguistic Landscape of Ciudad Juárez, Mexico

Natalia Mazzaro *, Natalia Minjarez Oppenheimer and Raquel González de Anda

Latin-US and Linguistics, College of Liberal Arts, University of Texas at El Paso, El Paso, TX 79968, USA; nminjarezo@miners.utep.edu (N.M.O.); raquelg@utep.edu (R.G.d.A.)
* Correspondence: nmazzaro@utep.edu

**Abstract:** Linguistic landscape (LL) studies in bilingual regions can reveal power dynamics between two languages, providing important information regarding their status and vitality. To analyze the relationship between Spanish and English in Ciudad Juárez, a city on the south side of the U.S.-Mexico border, we collected 1649 pictures of public signs in several sections of the city, whose "wellness levels" differ from each other. Pictures were coded for several factors, including language choice, business type, sign type, and the main and informative section, amongst others. Results show that while Spanish is the most frequently used language, English has a strong presence in the LL of Ciudad Juárez. The main factor affecting language choice is "business type". Certain businesses within the "beauty" category tend to favor the use of English, while businesses within the "home" category favor the use of Spanish. An analysis of socio-economic status (SES) and language choice revealed a direct relationship between them: English is favored in high-income neighborhoods, while Spanish is favored in low-income areas. The analysis of the main and informative sections on signs further confirmed the prestige assigned to English, which appears mostly in the main and most prominent sections of a sign. Our research shows that although Spanish vitality in Ciudad Juárez is strong, English is used in advertising because it is prestigious and increases the value of products and services, making them more appealing to shoppers.

**Keywords:** U.S.-Mexico border; advertising; marketing; language contact; Mexican commerce; linguistic landscape; Ciudad Juárez; English; Spanish; bilingualism

## 1. Introduction

Linguistic landscape (LL) research provides a visual representation of the multilingual nature of societies. By analyzing the visible language in public spaces, such as public road signs, store names, and advertisements, researchers can gain insights into the languages used by communities and the coexistence of different languages within a given area (Landry and Bourhis 1997). LL studies can also serve as a reflection of power dynamics and social hierarchies of languages within a community: the more visible or prominent a certain language is in the public domain, the higher its relative power and status in a speech community. Thus, LL studies can elucidate the distribution of power between co-existing languages, and shed light on issues pertaining to language dominance, language prestige, and the relationship between language choice and the social milieu.

The focus of this study is to investigate the LL of Ciudad Juárez, Mexico (a city bordering with El Paso, Tx. in the United States) and to understand the complex relationship between language use, prestige, and identity in the bilingual context of the U.S.-Mexico border. In a border city like Ciudad Juárez (CJ), it is common to see public signs in both Spanish, the dominant language, and English, the minority language. This is expected, given that CJ is adjacent to El Paso, Texas, where English is the majority language. An aerial view of the cities shows that CJ and El Paso (EP) form one big metropolis divided in two by a wall and a narrow river, the Rio Grande. Despite its adjacency to the U.S.-Mexico

border, CJ is a profusely Spanish monolingual community (Teschner 1995). The National Institute of Statistics and Geography (INEGI), which is Mexico's Census Bureau, focuses its language data collection on indigenous languages in Mexico rather than capturing information about all languages. According to the 2020 data, 0.56% of CJ's population speak an indigenous language and do not speak Spanish, which means that around 99% of *juarenses* do speak this language. In addition, a survey conducted with 1535 CJ residents by the El Paso Community Foundation (2018) revealed that 92.6% of respondents in CJ cannot speak English fluently, which suggests the population in CJ is predominantly Spanish monolingual. Yet, the presence of English can be clearly observed in signs around the city. This study investigates the social and linguistic factors that influence the use of English and bilingualism in public signage. There are a few studies that look at the relationship between Spanish and English in this part of the U.S.-Mexico border, and the most relevant and recent considered is the LL of El Paso, Texas (Mazzaro and González de Anda 2024). Other studies have analyzed *juarenses*' attitudes towards English, Spanish, and bilingualism (Hidalgo 1986) and young Mexican women's construction of identity through language use in social networks (Holguín Mendoza 2011). The present study will extend previous research by analyzing the presence of Spanish, English, and bilingualism in visual advertising and the factors that influence language choice. We base our study on the analysis of a collection of 1649 pictures of signs taken around CJ in January 2023, following our division of the city into different sectors according to wellness levels (WL), as will be explained in the methodology.

## 2. Literature Review

### 2.1. Linguistic Landscape

According to Landry and Bourhis (1997), LL studies are important because they can serve informative and symbolic functions. An informative function delineates the territorial borders of a language group and separates it from adjoining territories. Signs can indicate which language(s) is available for communication (e.g., when receiving a service or purchasing a product). On the other hand, the symbolic function of the LL relates to people's identification with a certain speech community (Gorter and Cenoz 2008), and the status and value given to the different languages spoken in it. This function has a more political and sociocultural motivation and can be related to feelings of language loyalty. By analyzing the LL, researchers can examine how "social actors" (Gorter and Cenoz 2008, p. 351) use language to attract customers, depending on the audience of the message and/or the type of business. Therefore, the findings of this study contribute important insights to sociolinguistics, marketing, and economics.

In addition to providing information about language status and power, LL studies can signal the sociolinguistic composition of the language groups who inhabit the territory in question. This is especially the case with private signs that are not regulated by the government, which more closely reflect the linguistic diversity of the population. Private signs are "commercial signs on storefronts and businesses, commercial advertising on billboards, and advertising signs in public transport and on private vehicles" (Landry and Bourhis 1997, p. 26), that are produced by actors that are generally members of the community and have knowledge of the community's linguistic code (Franco-Rodríguez 2009) or vernacular language. Conversely, public signs can be regulated by the government through language policy. These signs constitute "road signs, place names, street names, and inscriptions on government buildings including ministries, hospitals, universities, town halls, schools, metro stations and public parks" (Landry and Bourhis 1997, p. 26). While some countries, such as the U.S., do not have specific regulations for the language used in public signs, others, such as Mexico, do. Article 34 of Mexico's Federal Law of Consumer Protection states that the consumer information and labels of national and international products, as well as their advertising, should be in Spanish, and also indicate their cost should be expressed in the national currency. However, bilingual advertising, where Spanish is included alongside other languages, is permitted. In addition to the advertising of products, article 7 of the same federal law states that Spanish should be used

in the advertising of services and activities. In addition, article 22 of the Regulation of the Landscape and Urban Image of Ciudad Juárez mandates that signs should match the landscape of the area and be expressed in Spanish, except when the business was originally registered in a foreign language.

Corporate signs are different from private and public counterparts because the actors are not necessarily part of the community. This applies, for example, to franchises and international companies that have branches in different countries. Because of their global status, corporations may produce texts that are not always a reflection of the local vernacular (Cenoz and Gorter 2009). However, when corporations do include a local language in their signs, it can increase the in-group social prominence, because it recognizes their economic worth.

LL studies are also important because they provide insights into language vitality and language shift processes (Landry and Bourhis 1997)—that is, the presence or absence of languages in public spaces can indicate their strength or weakness in certain linguistic communities. When a language is present in the LL of a territory, it serves to stimulate its use, promoting language maintenance. Thus, the linguistic information offered by signs can serve to provide: (1) information about the status of a given language; and (2) implications for language policy, language planning, and language revitalization efforts.

Additionally, LL studies also offer information about the actors of a sign and the intended audience. A study by Spolsky and Cooper (1991) focused on the motivation for using some languages and not others on signs. They identified three rules that describe a sign, the author, and the intended audience: (i) the sign author writes signs in a language they know (i.e., according to the linguistic proficiency they possess); (ii) signs are written in the language(s) that the intended audience is assumed to know; and (iii) authors write signs in a language with which they wish to be identified. Rule (ii) has an economic motivation and is informative, while rule (iii) is symbolic.

Overall, LL studies contribute to our understanding of sociolinguistic dynamics, power relations, identity formation, and language policies within a given community. They provide valuable insights into aspects of linguistic prestige and reveal how attitudes towards different languages can influence language use.

The next section provides more information about CJ, including its population and geographical location, and also discusses the prestige of English, particularly in Mexico.

### 2.2. Ciudad Juárez Geography

CJ is located along the Rio Grande River on the US-Mexico border, south of EP, Texas. Despite not being the capital of the state, CJ is the most populated city in the state of Chihuahua, with 1,512,450 inhabitants (INEGI 2020). The city's initial population and industrial area developed along the railroad tracks and close to the border, which facilitated the export of manufactured goods (Gutiérrez Casas 2009).

The city has been growing continuously since the 1960s. Its urban sprawl was 16 times larger in 2005 than in 1960, and its population growth was, in the same time span, five times larger (Fuentes Flores 2008; Gutiérrez Casas 2009). Up until the 1960s, the city's downtown was considered the centralized employment area, where government offices, main financial institutions and businesses were located. The implementation of the *Programa Nacional Fronterizo* (National Border Program) and the *Programa de Industrialización Fronteriza* (Border Industrialization Program) by Mexico's federal government sought to improve the quality of life on the border by creating job opportunities. The accelerated expansion of urban sprawl and population growth resulted in a dispersed location pattern (Fuentes Flores 2008), and CJ transitioned from being a monocentric city to a polycentric one (Gutiérrez Casas 2009). Employment centers shifted from being highly concentrated in the north and downtown areas as a more spread-out pattern emerged, including some highly concentrated employment areas in the south (see Fuentes Flores 2008 for maps that show the geographical distribution of employment across time).

In addition, a combination of international and local factors resulted in the city's industrialization. For instance, in the period from the 1920s to the 1960s, the city's main industrial activity was cotton production. During WWII, the U.S. Government created the Bracero program, allowing Mexican agricultural workers to get temporary jobs in the United States. The Bracero program ended in 1964, which resulted in an unemployment crisis in the border city. To counter the employment crisis, the first industrial park and manufacturing plant (or maquiladora) was created. Maquiladoras attracted people from central and southern Mexico to the border (Castellanos 2018), and mainly included companies from the United States and Canada (INDEX 2023).

Although the maquiladora industry in CJ comes from many different countries, its employee population is mainly from Mexico and is Spanish monolingual (Teschner 1995). *juarenses* that do not have a way of crossing into the U.S. and lack the English language skills required to be upwardly mobile find lower paid employment in construction, janitorial and trucking jobs or, more precisely, in the maquiladoras (Teschner 1995). Thus, speaking English in CJ is essential because it improves workers' chances of getting a higher-paid job.

Unfortunately, Mexico's census does not, apart from indigenous languages, collect information about language skills, making it difficult to estimate how many people in CJ speak English. Private schools offer high-quality bilingual programs, but they tend to be a lot more expensive than public schools, meaning they are only available to the wealthy. Public schools at all levels offer English classes but they are very basic, thus bilingual education tends to be insufficient. This situation maintains social differences and contributes to the uneven distribution of economic resources among *juarenses*.

CJ's geographical adjacency to the United States causes it to have a highly dynamic population. Although most people who visit CJ from EP tend to stay for less than 24 h, they visit restaurants, casinos, motels, and souvenir shops (Gallegos and López López 2004). Gallegos and López López (2004) also point out that visitors from EP can be of two types. First, the ones who wish to visit a place with less harsh law enforcement, with more affordable goods and services, and with a weaker currency where their American salaries go further. Second, tourists visit CJ in search of business opportunities. According to U.S. Customs and Border Protection statistics (U.S. Customs and Border Protection 2023), 20.1 million vehicles and 5.2 million pedestrians crossed from CJ to EP in 2022. Many of these crossings were made by border commuters (i.e., people who live in CJ and work or study in EP, and vice-versa).

Going back and forth across the border is not a new practice. American tourism in CJ started in the early 20th century, when activities like gambling and drinking were prohibited in the U.S, causing related businesses to relocate south of the border. Several historical events affected touristic activities in CJ, as North American soldiers stationed in EP visited CJ for leisure, and the Bracero program attracted people from other places in Mexico to CJ. Although these events have long passed, there is still a high influx of people who arrive in CJ hoping to cross into the U.S and pursue the "American Dream".

Gallegos and López López (2004) list five main reasons for El Pasoan tourists to visit CJ: (1) nightlife, (2) restaurants, (3) souvenir shops, (4) downtown public markets and (5) medical services (optometrists, dentists, pharmacies, etc.). They also state that the touristic places are concentrated in Juárez Avenue, the street where the downtown international crossing is located. They categorize people who walk Avenida Juárez in three ways: (i) Mexicans on their way to cross to the U.S. downtown area for shopping or work; (ii) Americans who cross to CJ to go to restaurants, access medical services, or visit money exchange houses; and (iii) at night, people going to bars. The downtown port of entry seems to be the preferred international crossing because people can walk, not drive, which substantially reduces the time needed to go back to the U.S.

*2.3. English Prestige*

Because of American visitors and border commuters, English permeates the LL of Mexican commerce, meaning advertisements for international and local products are often

labeled with English names. While this linguistic influence can be partly attributed to language contact, which is a product of the geographic closeness of CJ and the United States, it is also indicative of a larger scale, worldwide trend that places English in a prestigious position (Baumgardner 2008).

Previous LL studies carried out in Mexico analyzed English use in public signage, and consistently found that English is viewed as superior and more modern (than the local language) in a given community. Baumgardner (2006) studied the use of English in Mexican businesses by interviewing managers from *Cigarrera La Moderna*, a boots and cigar manufacturing company located in Monterrey, Mexico. He found that English was chosen by store owners and vendors simply because "it sells". Further, the popularity of product names, such as Converse, Guess, and RayBan, is retained timelessly, as shown by this study, because imported brands are associated with a globalized product that is perceived as superior to the one produced in Mexico. This study supports the idea of English as an attention-getting device and as a marketing strategy, signaling that although members of the lower classes in Mexico may not understand the language used in advertisements, they are still attracted to products that use English in their branding (Baumgardner 2006). Although the international popularity of Spanish has grown, findings like these suggest that English is prestigious in Mexico because of its role as the world's lingua franca (Baumgardner 2006), and because of its frequent association with modernity, internationalism, and technological advancement (Gorter 2013). This is consistent with what Rosenbaum et al. (1977) more bluntly refer to as "snob appeal".

Other studies affirm English is a prestigious language, suggesting that socio-economic status (SES) intervenes with the degree of superiority ascribed to this language. Baumgardner (2008) analyzed Mexican newspapers from Monterrey, Mexico, and magazine advertisements from Mexico City to study how the use of English influences the perception of certain products. According to this study, individuals with higher SES are often bilingual, with the result that English is used to target higher income groups (Baumgardner 2008). For example, a comparison of the use of the words "fragrance" (English) and "fragancia" (Spanish) in perfume advertisements suggested that English use in Mexican products is not an attempt to fill a lexical gap in the host language, but instead reflects the advertiser's desire to use association with the English language to signal a higher status and quality that enhances the perceived value of products (Baumgardner 2008).

Studies of places in Mexico with high leisure tourism seem to replicate results from studies carried out in areas where tourism is industrial or business oriented, suggesting that English holds prestige in both touristic and industrial areas. Hoffman (2017) investigated the LL of Cozumel, Mexico, focusing on the signage and street advertising of pharmacies, finding that, although most pharmacies portray information in both Spanish and English, "the English denomination is significantly prioritized, as it is found bold in contrast to the signs of surrounding merchants" (Hoffman 2017). The ubiquitous presence of English across this island suggests a culture of globalized tourism where instances of English use represent group hierarchy and status (Hoffman 2017).

Together, these studies suggest that the status of English as a lingua franca across the international world remains, and that the degree of perceived prestige results in its prevalence across the LL of several non-English speaking communities. Therefore, it is not only geographical proximity to the United States that makes English prevalent in Mexican advertising and commerce, but also the degree of superiority this language seems to hold at the global level.

## 3. Research Questions and Hypotheses

We pose three research questions to examine the overall representation of English, Spanish, and bilingualism in CJ, and the influence of social and linguistic factors on the language choice of public signs:

1. How is the bilingual situation in CJ represented by signs?
2. What factors, linguistic and extra-linguistic, affect language choice in the LL of CJ?

3.  What do these findings imply in terms of language prestige and power in CJ?

The following predictions are made for each of the three research questions:

1.  We anticipate the presence of Spanish in the LL of CJ will be stronger than that of English, since it is the language spoken by the majority of the population (Teschner 1995). However, we expect English to be used as well, due to the prestige it holds in Mexico (Baumgardner 2006) and the proximity of CJ to EP. This prediction is supported by previous research focused on the LL of EP (Mazzaro and González de Anda 2024), which showed more Spanish is spoken closer to the border. We therefore expect a similar effect in those areas of CJ that are closer to EP.

2.  We predict "sign type" (corporate, public, or private) will influence language choice, with international corporate businesses using more English than national ones. Moreover, we expect that corporations that are binational (i.e., existing only in the United States and Mexico) will use bilingual Spanish-English signs to target customers on both sides of the border. Secondly, given that there are national and state laws that dictate Spanish as the language that should be used on signs, we anticipate public signs (i.e., official government signs) will have the highest percentage of Spanish use. Private signs, on the other hand, are expected to vary in language choice, since they are produced by actors that are generally members of the community, who can choose a particular language based on how they want their business to be perceived. Specifically, we anticipate higher rates of Spanish use in private and public signs compared to corporate ones, taking into consideration that this will also depend on the type of business (restaurant, health care, beauty, etc.), and not just on the type of sign (corporate, public, or private). For example, gyms in CJ tend to use more English than Spanish in their signs, irrespective of being corporate or private. In addition, we expect proximity to the U.S. border to promote a greater use of English in signs. This prediction is based on previous LL research of EP (Mazzaro and González de Anda 2024), which found more Spanish in signs closer to the U.S.-Mexico border. In the case of CJ, a higher English use of signs closer to the border would target pedestrians crossing from EP.

3.  Because of the status of English as a prestigious language in Mexico, we expect greater use of this language in signs located in areas with high SES. The status of a language can also be investigated through the analysis of the "main" and "informative" sections of signs (explained in Section 4.2). We hypothesize that English will appear more frequently in the "main" sections of signs that are more prominent, leaving Spanish to occur more frequently in the "informative" sections, which are less noticeable and provide additional information about the type of product and services offered by the business. Given that English is mainly used to assign prestige and/or status to a business, we believe that a minimal number of English signs will be translated into Spanish. Instead, we anticipate that signs will be English only or contain a section in English and another one in Spanish—for the purposes of this study, the term "code-switching" will be used when Spanish and English are used to convey different information within the same sign.

## 4. Materials and Methods

### 4.1. Division of Ciudad Juárez into Regions

Some cities such as EP have clear geographical divisions where major regions (e.g., west, east, downtown, border, northeast, etc.) are salient and separated by main highways, streets, or mountains. In cities like these, studying the characteristics of each individual region is possible, and allows for a clear-cut analysis. In EP, for instance, an analysis of the LL that follow these geographical divisions revealed a relationship between location, SES and language choice (Mazzaro and González de Anda 2024).

Unlike EP, CJ is not divided into clear geographical areas that dictate specific characteristics for each region. For instance, a generalization cannot be made about the east side of CJ, as this geographical area includes neighborhoods with high and low SES. To identify

if the SES-language choice relationship found in EP also exists in CJ, we approached this study in a different way. We followed a map developed by the Instituto Municipal de Investigación y Planeación (Institute of Urban Planning and Investigation of Ciudad Juárez, subsequently IMIP). This map shows divisions according to "wellness levels" (WL) that incorporate different sectors with the same economic, cultural, and social characteristics.

The divisions made by IMIP are based on the INEGI census data from 2020 (IMIP 2020). As shown in Table 1, the IMIP considered three groups of variables to account for the population's living conditions: (i) educational level (percentage of individuals with illiteracy and incomplete basic education, and average educational level attained), (ii) housing infrastructure (percentage of homes with dirt floors, lack of appropriate flooring, average number of rooms and bedrooms per home, etc.) and (iii) housing services (percentage of homes without running water, sewerage, electricity, a refrigerator, washing machines, and computers).

**Table 1.** Variables used by the IMIP to calculate WLs, based on information from the census (INEGI 2020). Our translation (to English) is in the right column.

| Variables | Variables |
| --- | --- |
| *% de Población Analfabeta* | % of Illiterate Population |
| *% de población con rezago educativo* | % of population with educational lagging |
| *Grado promedio de escolaridad* | Average schooling |
| *Promedio de ocupantes por vivienda* | Average number of people per household |
| *Promedio de ocupantes por cuarto* | Average number of people per room in the household |
| *% de viviendas con piso de tierra* | % of households with dirt floors |
| *% de viviendas con un dormitorio* | % of households with one bedroom |
| *% de viviendas con 2 o más dormitorios* | % of households with two bedrooms or more |
| *% de viviendas con un cuarto* | % of households with one room |
| *% de viviendas con dos cuartos* | % of households with two rooms |
| *% de viviendas con tres cuartos y más* | % of households with three rooms or more |
| *% de viviendas que no disponen de servicio sanitario* | % of households without sanitary services |
| *% de viviendas que no disponen de agua entubada en el ámbito de la vivienda* | % of households without running water |
| *% de viviendas que no disponen de drenaje* | % of households without sewage |
| *% de viviendas que no disponen de energía eléctrica* | % of households without electricity |
| *% de viviendas que nodisponen de servicios completos* | % of households without essential utilities |
| *% de viviendas que no disponen de refrigerador* | % of households without a refrigerator |
| *% de viviendas que no disponen de lavadora* | % of households without a washing machine |
| *% de viviendas que no disponen de computadora* | % of households without a computer |

A combination of the three groups of variables described above (education level, housing infrastructure and housing services) results in five different WLs that are represented through color-coding: dark green (very high WL), light green (high WL), light yellow (medium WL), dark yellow (low WL), and red (very low WL). Figure 1 shows the color-coded divisions scattered throughout the city. While an initial glance suggests that colors are scattered randomly across the map, closer inspection reveals a pattern that is worth mentioning: the yellowish areas (medium and low WL) are in the outskirts of the city, the red areas (very low WL) are in more peripheral locations, and the green areas are in the north-central, central, and south-central parts. This may be taken to suggest that lower socio-economic areas tend to be pushed towards the peripheries of the city.

For our study, we took pictures of the public signage in commercial and industrial (not residential) areas of each color. To obtain a sample that was representative of CJ, we took pictures from at least two sectors labeled with the same color. From each of these sections, we selected busy areas, including main streets with heavy traffic and a significant number of pedestrians. The samples were equivalent because they all included a varied distribution of business types. For instance, all areas included pictures of each category: food, religion, clothing and apparel, etc.

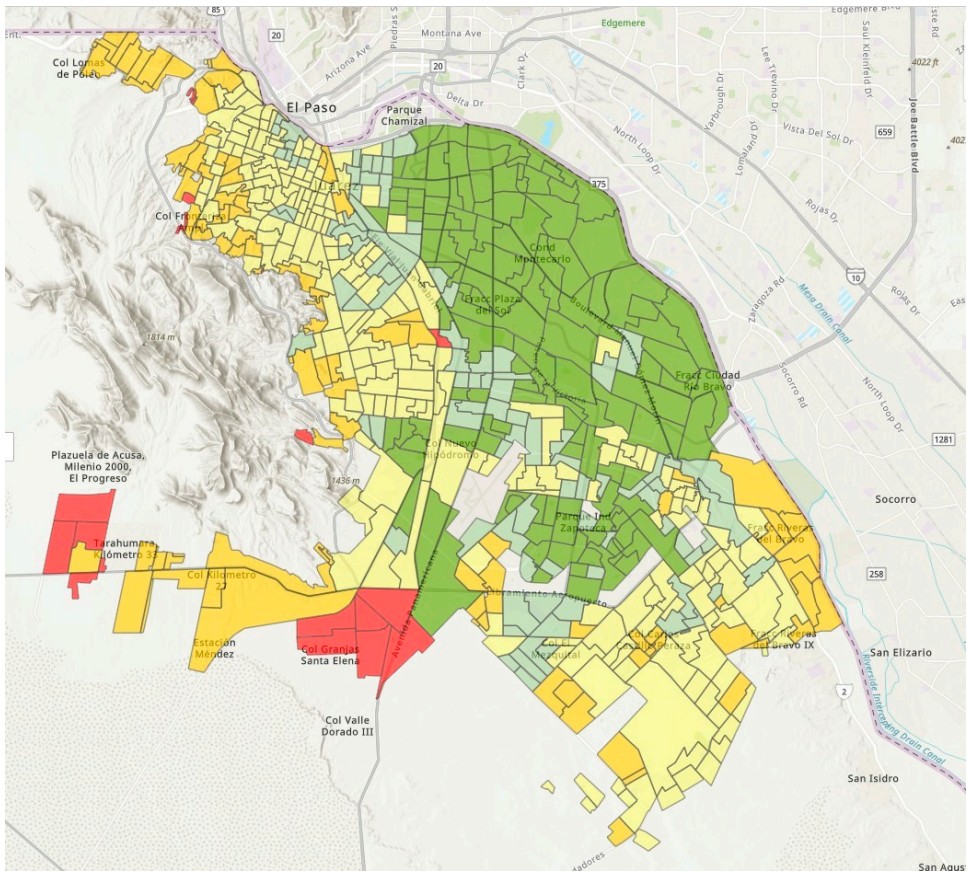

**Figure 1.** Division of CJ by "wellness level" (WL), as categorized by the IMIP. The colors represent different WLs: dark green (very high WL), light green (high WL), light yellow (medium WL), dark yellow (low WL), and red (very low WL).

*4.2. Data Collection and Factors Analyzed*

All the pictures were taken with the authors' smartphones during three different visits to CJ in January 2023. All authors were present during each session, as this avoided the same pictures being retaken. Different sections of the city were visited in accordance with the map division developed by the IMIP, with the aim of taking at least 200 pictures from each of the five WL sectors. Pictures of all WL (except for the red WL) were taken from the sidewalk. For safety reasons, all the pictures from the red WL were taken from the car. While some studies aim to take pictures from a similar number of blocks per area, we instead based our analysis on a similar number of pictures per area, as we believe this is preferable in cases of cities that have an uneven urban distribution, such as CJ. According to Pérez Pulido and Romo Aguilar (2022), the rapid expansion of the population has resulted in poor urban planning in CJ, which has been exacerbated by the local government often ignoring IMIP's recommended improvements.

Gorter (2006) observes that what is considered the "unit of study" can vary across LL studies and asserts that it must be defined. For this study, we only analyzed pictures of signs with text "within a spatially definable frame" (Backhaus 2006) that are not placed on a moving object, such as a car or bus. Also following Backhaus (2006), we included signs of very different sizes, from small posters on store fronts to billboards. We excluded signs that were not big enough to be read from the sidewalk.

Our methodology is partly based on techniques employed by previous LL studies, such as Carr's (2021) research of the LL of Los Angeles. Carr (2021) stressed the importance of analyzing public signage by separating main and informative sections within a sign, and this methodological approach has also been endorsed by Franco-Rodríguez (2008, 2009). The main section of a sign is its most salient part, usually containing a larger-sized

font. The language choice of this main section can reflect a language's prestige, relative to other languages in the area (Franco-Rodríguez 2008, 2009). The informative section of a sign, on the other hand, is considered secondary and contains additional information that complements what is found in the main section, and its font size and colors are usually more conservative (Carr 2021). According to Franco-Rodríguez (2008, 2009), this informative section can often display information in a minority language, representing a useful way of communicating with the local community. Hence, the fact that a language can be presented in a larger and more salient way than another language sheds light on the power dynamics within a community's language choices and attitudes.

As will be seen in the coding categories section below, our study implements the technique proposed by Franco-Rodríguez (2008, 2009) and utilized by Carr (2021), in which the content within a sign's main and informative sections are analyzed separately for language choice.

An Excel file was developed by the authors as a coding mechanism for each sign. This spreadsheet contained different columns with categories that served to describe the characteristics of each sign. The first category in the coding spreadsheet was "languages". Signs in which the text was all in Spanish, or included English borrowings, were coded as "Spanish". On the other hand, signs with all-English text, or with some Spanish borrowings, were coded as "English". Signs that used both languages (e.g., using code-switching instead of borrowings) were labeled as "bilingual". Since the focus of this study is the use of Spanish and English in the region, signs with text in a language that was neither Spanish nor English, or that included non-sense words or brand names like "OXXO" (a company chain that owns numerous convenience stores across Mexico), were coded as "other language that is neither English nor Spanish".

The second category analyzed was "location". Here, the aim was to investigate the relationship between SES and English use. Using the map in Figure 1, we coded each picture according to the color of the WL where the picture was taken. In addition, a third category coded for closeness to the border, labeling signs as "border" or "non-border", depending on where they had been taken. Closeness was defined as the space between the *Santa Fe* international port of entry and the *16 de Septiembre* street: pictures coded as "border" were taken between these two points, while those taken outside this perimeter were coded as "non-border". This street was chosen as a limit to define closeness due to its heavy pedestrian traffic and because it is the only one in the city that features a commercial area that is adjacent to the border and is accessible for pedestrian border-crossers. Mazzaro and González de Anda (2024) found that in EP, Spanish was more frequently used on signs located near the border, as businesses wish to attract shoppers from CJ. Coding for this information in CJ will show us if the same phenomenon is observed when we get closer to the border with the U.S.

The fourth coding category ("sign type") provides information about the sign creator. A sign coded as "public" was created by the government (e.g., signs in parks and government offices, or on monuments). On the other hand, signs coded as "private" were created by local businesses and individuals, and those coded as "corporate" by franchises. Businesses classified as "corporate" were further distinguished into "national" (e.g., Telcel), "international" (e.g., McDonald's), or "binational" (those that only exist in Mexico and the United States; e.g., Taco Tote). The differentiation of public, private, and corporate signs is important because language choices made by the authorities and the citizens are not always the same (Backhaus 2006), and because locals and non-locals may make different languages choices (Franco Rodríguez 2013) that impact the LL of a particular community.

Our fifth category ("business type") analyzed the different types of businesses in CJ. This category is important because, as stated earlier, people from EP who visit CJ do so for specific purposes (Gallegos and López López 2004), meaning that different businesses can have a varying influence on language choice. The business types included in this study, which are based on Dings and Hertel (2015) and Mazzaro and González de Anda (2024), are food, vehicle, beauty, education, hobbies and entertainment, religion, restaurant and

catering, health care, legal services, clothing and apparel, home, financial, communication, office supplies, miscellaneous and travel. Each of these business types included a number of specific business types, and a note was made that specified in detail which business a sign pertained to. For example, the business type "beauty" included specific businesses such as hairdressers, weight loss centers, beauty clinics, nail salons and laser hair removal.

The subsequent coding categories helped us differentiate between the main and informative sections of signs. As suggested by Franco Rodríguez (2013, p. 113), "[a]n analysis that correlates language placement with text content can provide information about the symbolic and informative functions of a language". For our study, we coded the language used in the main and informative sections separately. Each section was coded as "Spanish", "English", "Both" (English and Spanish), "other", or "none". "None" was used in cases where, for example, a sign did not include an informative section.

The next coding category was "translation" and only pertained to bilingual signs. Signs with text in both languages were coded as "word for word translation", which applied when the same information was presented in English and Spanish (e.g., *Se vende*—For sale). Alternatively, signs were coded as "no translation" when they had different information in English and in Spanish (e.g., *Reconocidos 18 años como* un great place to work. "Recognized for 18 years as a great place to work"). Lastly, signs were coded as "partial translation" when only some of the information was translated to the other language. For example, "*Farmacia* Drugstore, *más ofertas todos los días*" (Drugstore. More sales everyday).

The last coding category was related to linguistics standards, and the codes were "standard", "code-switching", "borrowings" and "orthographic errors". In this category, we focused on the use of code-switching, since the main focus of this study is the contact situation between Spanish and English. Thus, if a sign presented both orthographic errors and code-switching, preference was given to code-switching, and the sign was coded as such. For example, one sign featured the following text, which included both orthographic errors (the lack of a stress mark in "*dulcería*"/"candy store") and code-switching ("Candy land. *Dulceria y mas*"/"Candy land. Candy store and more") (see Figure 2).

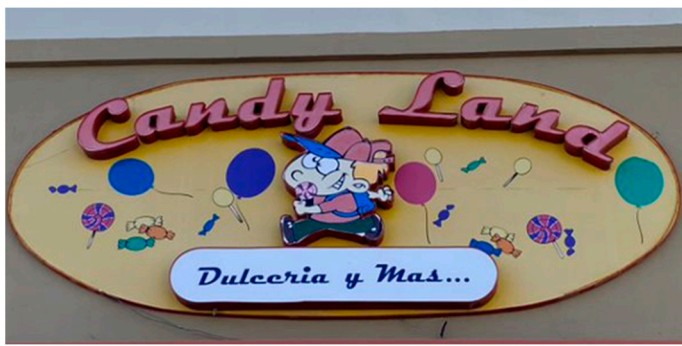

**Figure 2.** Bilingual sign with a main section in English, ("Candy Land"), and an informative section in Spanish, "*Dulceria y mas*"/"Candy store and more".

The sign in Figure 2 was coded as code-switching, and not as orthographic error. However, in a future analysis, we will add another column that adds a second label to tokens that include a combination of linguistic standards.

### 4.3. Coding and Analysis

Our initial analysis began by coding 50 pictures from each of the 5 color-coded regions. This was done with the purpose of: (i) obtaining a set of preliminary results that would allow us to observe any initial patterns in the data (e.g., more English use in certain areas of the city), (ii) test the effectiveness of our coding procedure, and (iii) determine how user friendly the Excel spreadsheet would be for the students who would help us code the remaining pictures.

After our preliminary analysis of 250 pictures, students from an upper division Spanish course at a university located on the U.S.-Mexico border received two 80-min training sessions from the researchers, in which they were taught how to code pictures on the Excel spreadsheet. Each of the 20 students received a set of 35 pictures to code, yielding a total of 700 pictures. Besides receiving class credit for their participation in this activity, students used this activity to practice for their final project, where they had to develop a LL study of their own. The remaining 699 pictures were coded by the researchers, and all 700 student-coded pictures were subsequently subjected to a reliability check, where the researchers reviewed each coding for correctness.

Our results use a combination of cross-tabulations with chi-square analyses, using Jamovi (The Jamovi Project 2023), Random Forest and Variable Importance analyses (Strobl et al. 2008) that apply R (R Core Team 2020).

Cross-tabulations were performed to determine the distribution of the dependent variable (language choice) across each independent factor: sign type, location, business type, specific business. Chi-square analyses were conducted on each of these tables to evaluate if a given independent factor had a significant effect on the dependent variable. Random Forest and Variable Importance analyses are important because they help determine the relative ranking of factors against each other; in other words, they tell us which factor(s) has more influence on the dependent variable than others. Random Forest creates many random subsets of the data, and a conditional inference tree for each subset. The conditional inference tree determines a series of binary splits, starting with the strongest predictive variable and working down until no predictor is statistically significant. The Variable Importance analysis then looks at the results of each of these trees to see which are the most important overall predictors. Running many random subsets of the data (as provided by the Random Forest) adds robustness to the overall analysis. Finally, we conducted several binomial regression analyses, adding the factors one by one in the order of the Variable Importance analysis to determine which factors improved the fit of the model to the data. Together, all these analyses allowed us to evaluate which independent factor was more important in predicting language choice in the signage of CJ.

## 5. Results

The first question investigated the overall occurrence of Spanish, English and Spanish-English bilingualism in signs in CJ. Our results show that Spanish has the highest percentage of use (63.1%), followed by bilingual signs (28%) and English (7.5%). The category with the least number of tokens was "other" (1.5%), which included languages that are neither Spanish nor English. These results confirm our hypotheses, which predicted that while both languages, Spanish and English, will be present in the LL of CJ, Spanish will be stronger.

The second question investigated the individual factors that are hypothesized to influence language choice in the signs of CJ. Starting with "sign type", corporate signs were found to have significantly higher rates of English (11.3%) than private (6.8%) and public (2.6%) ones (Figure 2). The use of Spanish presents the exact opposite effect, with significantly higher frequencies of Spanish use in public signs (89.7%) compared to private (63.5%) and corporate (54.9%) counterparts. Figure 2 shows how the rates for English and bilingual signs decrease from corporate to public signs and how Spanish increases from corporate to public signs. A chi-square analysis shows significant differences in the use of Spanish, English and bilingualism across sign types $\chi 2$ (6, n = 1649) = 54.4, $p < 0.001$. To determine which categories were significantly different from each other, we compared each pair of categories. We did this by building a $2 \times 3$ contingency table for each pair and calculated a chi-square by using a Bonferroni correction (i.e., dividing the confidence level of 0.05 by the number of comparisons (6) to obtain a new limit for significance (0.00833)). These pairwise comparisons showed significant differences between public corporations and the other two counterparts (Figure 3). No significant differences were observed between corporate and private corporations, which may indicate that these categories are composed of other subcategories that are worth analyzing.

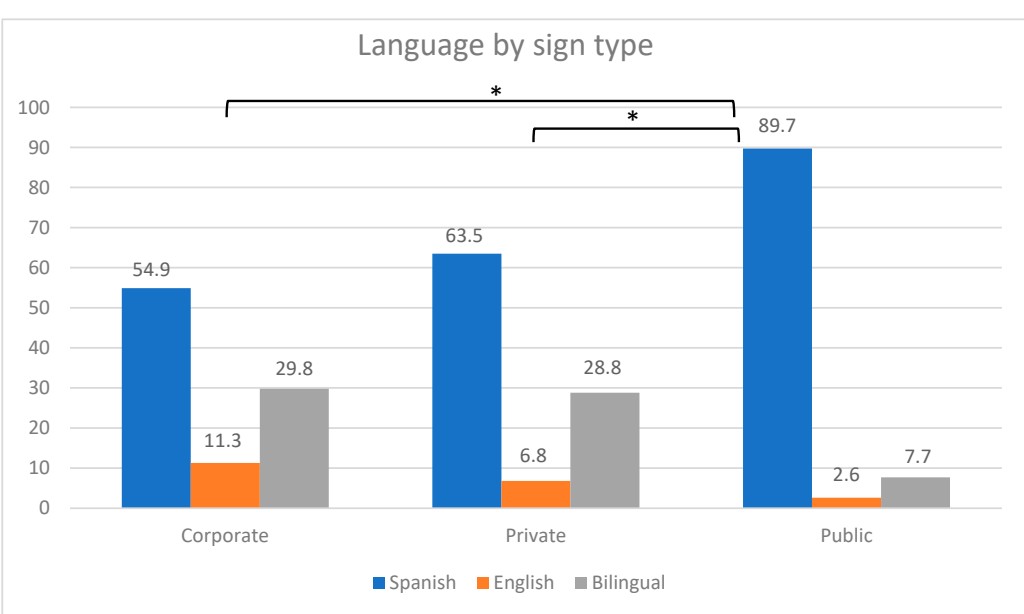

**Figure 3.** Distribution of Spanish, English and bilingual signs across sign types. The asterisk symbol (*) indicates a significant difference.

Taking a closer look at the corporation category and dividing the tokens into "international", "binational" and "national" showed a significant difference in their behavior ($\chi 2$ (6, n = 318) = 43.6, $p < 0.001$). The results in Figure 4 show that international corporations have larger proportions of English use (23.5%) than binational (17.6%) and national (7.3%) counterparts. The opposite effect is found for Spanish, where national corporations have higher rates of Spanish use (63.9%) than binational (52.9%) and international (23.5%) counterparts. Pairwise comparisons showed significant differences between international and national corporations.

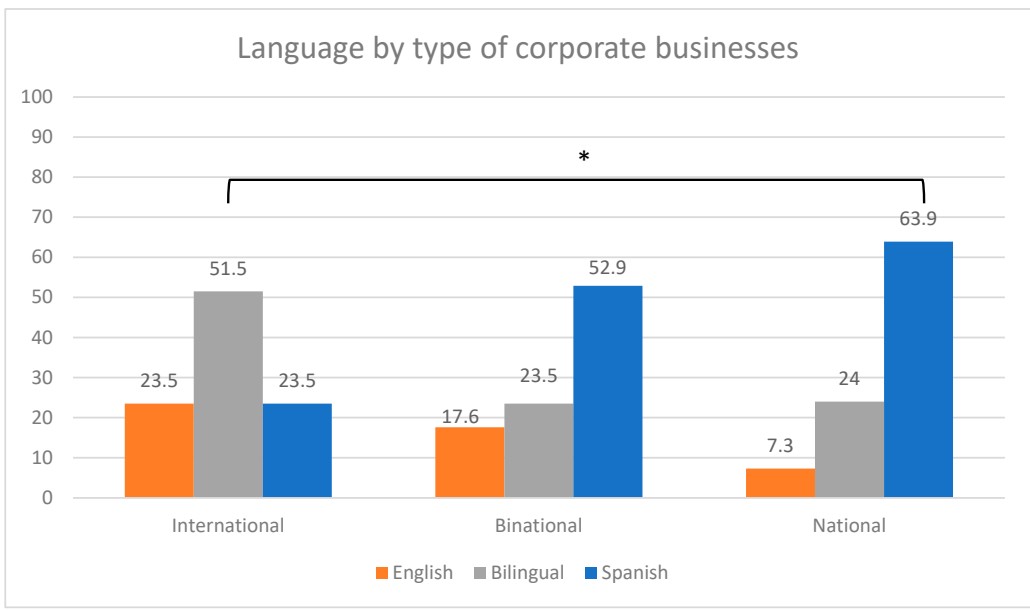

**Figure 4.** Distribution of Spanish, English and bilingual signs across different types of corporate businesses: "national", "binational" and "international". The asterisk symbol (*) indicates a significant difference.

We also analyzed the language choice in the signs of private businesses, according to business type, as shown in Figure 5. This analysis excludes the category "other language", which had very few tokens overall (n = 25), leaving us with 1624 tokens for this analysis.

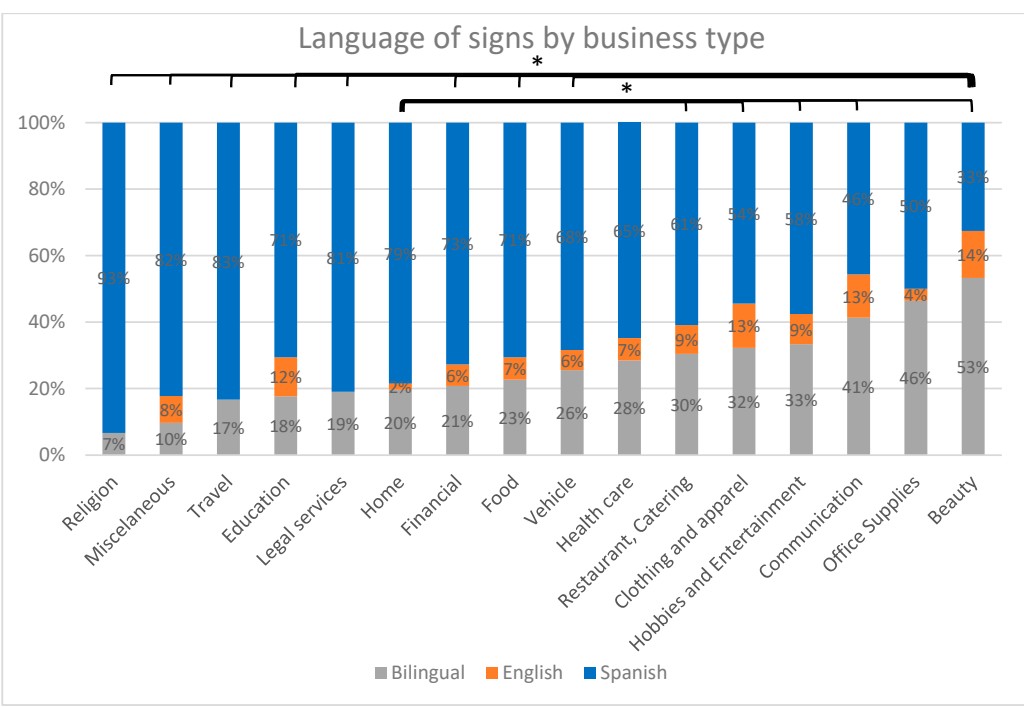

**Figure 5.** Distribution of Spanish, English and bilingual signs across different business types. The asterisk symbol (*) indicates a significant difference.

In this section we discuss language choice by business category by referring to the bilingual context at the U.S.-Mexican border. To make the comparison easier to observe, we ordered categories according to percentage of English use in signs. English and bilingualism were collapsed for this analysis because we are interested in signs that have English, relative to those that do not. Starting on the right, we see that there is a high percentage of English + bilingualism in "beauty" signs (67.4%). Beauty services are a main attraction to CJ from EP, because they offer the same services at a fraction of the U.S cost. In fact, Mazzaro and González de Anda's (2024) LL study of EP shows that the beauty category accounted for one of the highest percentages of English in signs (77%), leading them to observe that "[b]eauty services are less costly in Juárez, so these businesses in El Paso probably target (English speaking) locals almost exclusively". The high proportion of English in this category could also be related to the prestige that English holds in Mexico (Baumgardner 2008), where the language is used to signal higher status and quality, along with sophistication and modernity. These are all characteristics that beauty products and services may want to convey by using English in their signs.

The next category with a high percentage of English and bilingualism is "office supplies", which includes a high proportion of bilingual signs with words such as "print", "full color", "flyers" and "signs". One possible explanation for this trend of using English words in otherwise Spanish signs could be directly taken from the aforementioned "beauty" category: English is used as a strategy to signal product superiority, higher quality and sophistication: a business advertising "full color prints" is therefore perceived to be more prestigious than a counterpart advertising the same product in Spanish ("*impresiones a todo color*"). The high percentage of English use in this business category may not necessarily be concerned with attracting El Pasoans, but rather with signaling higher quality through the use of a particular language.

The category "communication", which includes telecommunication services and products, internet providers, phone carriers, satellite TV, streaming services, and cellphone repair, is next, in terms of percent of English and bilingualism (54%). Here, the high proportion of English use could be due to the influence of English in the technology lexicon: many words related to technology are already borrowed into Spanish (internet, streaming, iPhone, software, etc.), but many signs included English words that are not part of the Spanish lexicon, such as "laptop doctor", "software factory", "family cellular", "total play", "compu and games", etc. However, there are other possible explanations found in the literature. According to a report by Nielsen in 2012, the companies that produced most "Spanglish" advertisements are Dish Network, AT&T, Verizon, Walmart, McDonald's, General Mills, Kraft Foods, Toyota and General Motors (Escobar and Potowski 2015). The first three in the list are part of the communications category, suggesting a trend to use both Spanish and English to advertise to Latin American customers. Also, as shown earlier, international companies tend to advertise in English, meaning that all these aspects could be interacting to strengthen the presence of English in this category.

The category "hobbies and entertainment" also had a high rate of English and bilingualism in signs (42.4%), which corroborates the claim by Gallegos and López López (2004) that one of the main reasons why El Pasoans visit CJ is for entertainment and nightlife options. We believe that some businesses within the entertainment domain cater to border commuters, who take advantage of more affordable prices on the Mexican side of the border. These businesses include nightclubs, movie theatres, photography services, party supply stores, and also gyms, who overwhelmingly prefer to use English in their signs.

"Clothing and apparel" also contained a high proportion of bilingual and English signs (45.6%). Since many people from the state of Chihuahua cross to EP to shop for high-end brand shoes, clothing and bags, we believe that the use of English in this category is due to the status associated with English. As proposed by Baumgardner (2008), the prestige of English is associated with SES, since many Mexicans who are bilingual are also from high SES. Thus, the use of English is a medium used to add superiority to the product, and to simultaneously target higher income groups (Baumgardner 2008): for Mexican shoppers, it may be 'cooler' and more attractive to buy a pair of jeans from an outlet jeans store than from one that offers *pantalones de mezclilla baratos*.

Another category with high rates of bilingual and English signs (39.1%) is "restaurants and catering". In addition to restaurants, this category also includes bars, coffee shops, food trucks, home-based food businesses, and snack shops. Within this category, bars account for the highest percent of use of English in their signs (51%). We believe that many of them could be catering to the EP community, who seek to take advantage of less costly opportunities for entertainment—this recalls the category "hobbies and entertainment", which suggested nightlife is one of the main reasons why EP tourists visit CJ. Gallegos and López López (2004) proposed that CJ's excellent restaurants, which offer outstanding food at a lower price than EP, are another reason to visit. In addition to targeting EP visitors, restaurants and bars may also use English in their signs to appear Americanized and to look "worldly" and "cool".

The final category to frequently use English in signs is "health care" (35.2%), a large category that grouped several specific businesses, such as dentist offices, veterinarians, optical services, pharmacies, clinics, medical centers, hospitals, and medical supplies. Of these businesses, the first three accounted for the most data, so we will focus our discussion on them. Dentist offices are one of the main businesses in CJ that cater to EP residents. Many El Pasoans who do not have medical coverage or who do not want to pay the exorbitant cost of medical care in the U.S cross to CJ to receive these services at a fraction of the cost, as illustrated by the large proportion of English signs advertising dentist offices (46%), veterinarians (50%), and optical services (35%). Thus, health tourism is another reason why El Pasoans cross to CJ (Gallegos and López López 2004), as our data demonstrates.

The rest of the categories appear to be Spanish dominant, but some (education, legal services, religion and travel) had very few tokens, so we will focus our discussion on

"home", "food" and "vehicle". The category "home" included a variety of sub-categories, such as construction supplies, repair shops, decor stores, appliances, key shop, utilities, and others. These services yielded a high percentage of Spanish use (79%), suggesting that the signage mainly addresses the local community. Within the category "food", which includes grocery shopping, convenience stores, snack shops and other food products, a high rate of Spanish use was also found (71%), supporting the idea that these products mainly cater to *juarenses*. Lastly, the category "vehicle", which grouped together several businesses like repair shops, car dealerships, car wash, auto parts stores, public transportation, parking and gas stations, also showed a large proportion of Spanish-only signs (68%). Once again, this finding suggests that services related to vehicles are targeted at city dwellers. A careful look at the signs shows that many repair shops included in our data are small private stores with rather informal, and often handwritten signs, and it is unclear whether the high percentage of Spanish is due to a willingness to target locals or a lack of English proficiency.

To determine which categories were significantly different from each other, we compared all pairs of business types. There was a total of 16 business types, so we obtained $16 \times 15 = 240$ pairs. We built a $2 \times 2$ contingency table for each pair and calculated a chi-square for each, using a Bonferroni correction (i.e., dividing the confidence level of 0.05 by the number of comparisons (240) to obtain a new limit for significance (0.000208)). Pairwise comparisons showed significant differences between "beauty" and any one of "vehicle", "food", "financial", "legal services", "education", "travel", "miscellaneous", and "religion". Significant differences were also found between "home" and any one of "communication", "hobbies and entertainment", "clothing and apparel", "restaurant and catering" and "beauty". Despite having high rates of English and bilingualism use, "office supplies" did not reach a significance level, due to having less tokens than the others (Figure 4).

We also investigated if businesses adjacent to the U.S. border would show greater use of English in signs, and this prediction was based on the previous LL research of EP (Mazzaro and González de Anda 2024), which found more Spanish in signs closer to the border. We expected that the same effect would be seen in CJ, and that there would be a higher percentage of English use closer to the border with the U.S. However, contrary to our expectations, the chi-square analysis showed that differences were not significant ($\chi^2$ (3, n = 1649) = 4.23, *p* = 0.237). Although there were more bilingual signs closer to the border (border 32.8% vs. non-border 27.3%), the percentage of Spanish and English signs was slightly higher in non-border areas (Spanish: border 60.6% vs. non-border 63.4%; English: border 6.1% vs. non-border 7.6%), which suggests that border crossers coming to CJ are not, in most cases, pedestrians, and instead seek to drive to specific locations in CJ, regardless of the distance from the border, in order to look for specific services or businesses.

Our next analysis looked at the relationship between SES and language choice in signs. Figure 6 illustrates the distribution of languages across areas with different WL. The distribution shows a very clear pattern of higher percentages of English use in areas of high WL and higher rates of Spanish use in areas with low WL. A chi-square analysis shows significant differences in the use of Spanish, English and bilingualism across areas ($\chi^2$ (12, n = 1649) = 69, *p* < 0.001). Pairwise comparisons showed significant differences between the "very low" SES group and the rest of the categories. There were also significant differences between the "very high" SES group and the next two groups in the social ladder ("high" and "medium" SES). These results confirm our hypothesis that the use of English is related to SES; accordingly, when SES is higher, so is English use.

To analyze the relative importance of factors that affect the occurrence of English and bilingualism in signage, we conducted a Random Forest and Variable Importance analysis (Strobl et al. 2008). The results are shown in Figure 7.

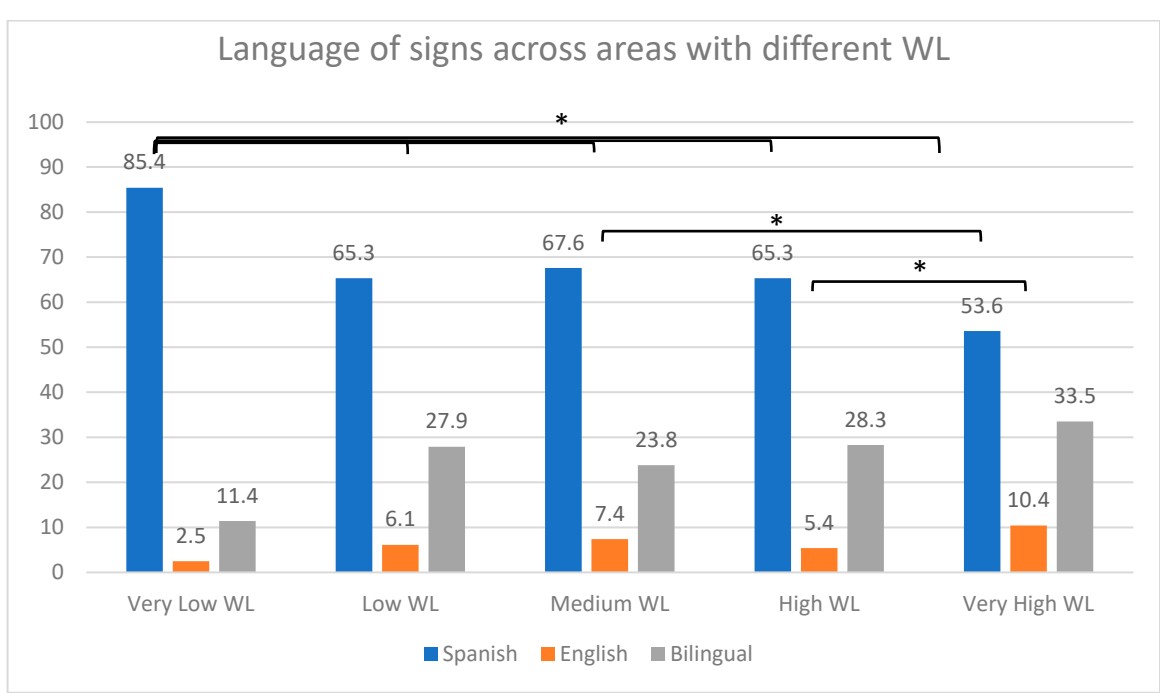

**Figure 6.** Distribution of English, Spanish, bilingual and other language (neither English nor Spanish) signs across areas in CJ. The asterisk symbol (*) indicates a significant difference.

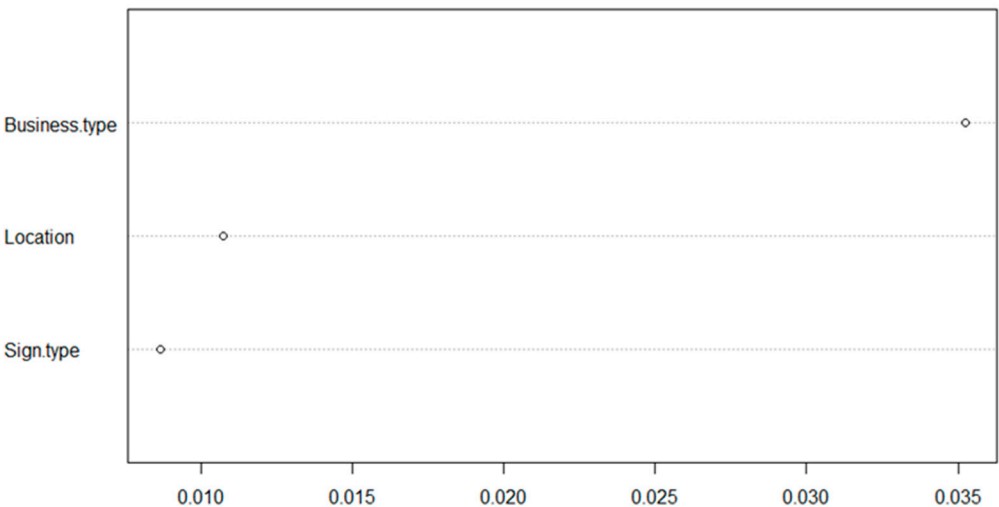

**Figure 7.** Random Forest with location, business type and sign type.

The Random Forest shows that business type is the most important factor in predicting language choice in signs, followed by location and sign type. To determine if all three factors are important in predicting language choice, we conducted a set of binary logistic regression analyses, where the three factors were added one by one to the model, and the AIC and BIC were compared after each addition. AIC and BIC are measures of goodness of fit (of the model to the data), and a lower AIC and BIC indicate a better fit. A difference of two points or more in AIC and BIC between models indicates that they are significantly different. The results showed that all the factors (business type, location, and sign type) contributed to the improvement of the model, which means that they are all important predictors of language choice in signs.

To address question (3) regarding the status and/or prestige of English in the LL of CJ, we looked at several factors. First, as already shown (Figure 6), the analysis of signs across areas with different WLs showed that there was a significantly greater use of English and

bilingualism in areas with higher SES. The preference of English by members with higher SES may indicate the prestige assigned to the language.

To further corroborate the relationship between language choice and prestige, we analyzed the different sections of signs (i.e., main and informative). The main section of a sign is associated with symbolic power, so we anticipated that the use of English would be higher in this section than in the informative section, which describes the service and product advertised. To perform this analysis, we only used signs that contained both main and informative sections (n = 1235). The results are shown in Figure 8.

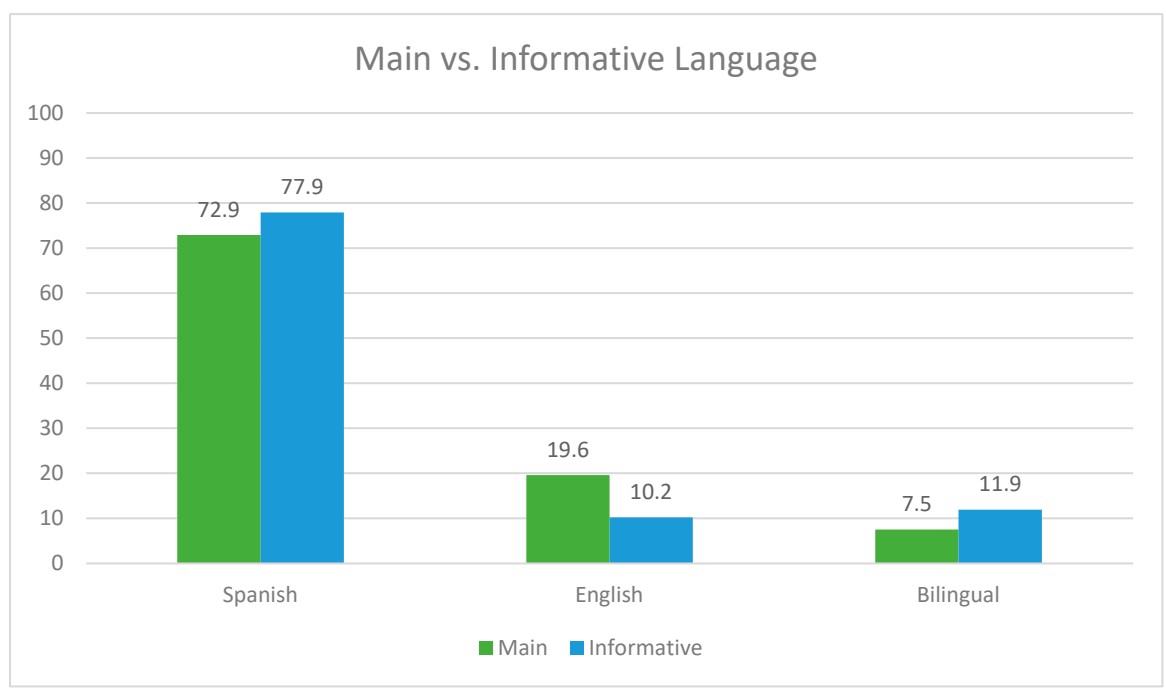

**Figure 8.** Distribution of Spanish, English and bilingual texts in main and informative sections.

As predicted, the greatest difference between main and informative sections is within English. There is a higher rate of English in the main section (19.6%), compared to the informative counterpart (10.2%). Conversely, there was a higher proportion of Spanish in the informative section (77.9%) than in the main section (72.9%). Bilingual informative sections were also more frequent (11.9%) than bilingual main sections (7.5). Signs such as the one shown in Figure 2 illustrate this typical pattern of bilingual signs in more detail.

Finally, we analyzed the proportion of English-containing signs that offered translation into Spanish. This analysis included signs containing English, whether bilingual (n = 461) or English only (n = 123). Our results show that the large majority of signs do not offer any translation (93.8%). Partial translation was given in a small percentage of signs (4.6%) and word by word translation was only observed in 1.6% of the signs. This pattern reveals that the use of English is not informational but is instead symbolic. Rather than addressing an English-speaking population, signs containing English are used to attribute prestige to a business or product.

To complete our analysis, we investigated how English is incorporated into an otherwise Spanish discourse (as in the category "linguistic standards"). For the analysis in Figure 9, we only considered Spanish and bilingual signs (n = 1501 tokens).

Figure 9 shows that Spanish only signs are mostly standard (80.6%), with a low percentage of orthographic errors (6.8%) and slang (3.1%). The high rate of standard Spanish suggests that its vitality remains intact, despite the strong presence of English in the signage. In addition, the percentage of English borrowings into Spanish was 9.5%, suggesting that when English is incorporated in signs it is mostly in the form of code-switching (95.5%), as shown in the bilingual portion of Figure 8.

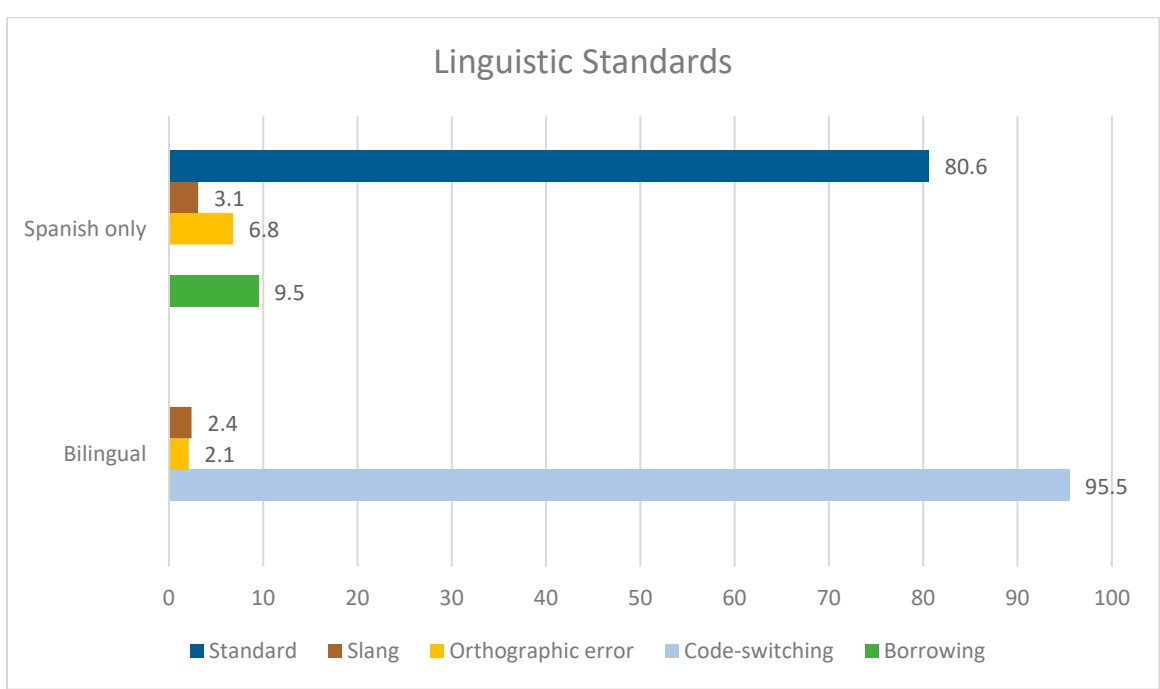

**Figure 9.** Distribution of Spanish only and bilingual signs across linguistic standards.

## 6. Discussion and Conclusions

This section is structured in order to address our research questions and hypotheses.

The first general question investigated the linguistic representation of Spanish, English, and bilingual signs in the LL of CJ. We also coded and analyzed other languages present in signs around the city (including European languages such as French and Italian, and Asian languages like Japanese and Korean), but their proportion was so low (1.5%) that we excluded them from further analysis. In addition, this group also featured businesses whose names were made-up nonsense words, such as "izzi" or "S4H". The low representation of languages other than English and Spanish in signs is informative, as it matches the demographic composition of CJ, where the majority of the population are Mexican nationals (91%) and the rest are international immigrants, many of whom seek the U.S. as their final destination (INEGI 2020).

Our first hypothesis about the strong presence of Spanish and English in the LL of CJ was therefore supported. Our results demonstrate that, while most signs are in Spanish, English is used in the main section of many signs, giving it the most prominent position. We therefore argue that English is used to convey prestige and to attract border crossers from the U.S who visit CJ for specific purposes. These results are consistent with the findings of Gallegos and López López (2004) and the "citizen perception survey" conducted by the El Paso Community Foundation (2018), which states that the four main reasons why El Pasoans cross to CJ are visiting friends/family, shopping, doctor/dentist appointment/visiting drug stores and entertainment.

The second question explored the factors hypothesized to affect language choice in the LL of CJ. Of all the factors considered, the type of business had the strongest effect on language choice. Certain categories, such as "beauty" services, promoted the use of English more than others. Other businesses favoring the use of English were "communication", "restaurants and catering", "hobbies and entertainment", and "health care".

The high rate of English in the signs of these businesses may be motivated by two goals: (i) to attract border crossers who visit CJ for specific services; and (ii) to add prestige to the product or service advertised. Other businesses within the "home" and "vehicle" categories had significantly higher rates of Spanish use. These categories encompassed repair services as well as products, and so they were mostly targeting the local community and advertising in Spanish. While we anticipated that type of business would influence

language choice, we did not expect it to be the most important predictor. These results show that the economic motivations for using English surpass the symbolic ones: while all businesses could potentially use English to add prestige to their service or product, businesses like beauty salons, which show higher rates of English use in their signage, may be doing it for a combination of reasons, including those that originated in both practical (e.g., attracting customers from across the border) and symbolic (e.g., to add prestige to their business) motivations.

Our analysis of corporate and private businesses showed that looking at these categories in more detail provided a more accurate picture of the factors affecting language choice in the LL of CJ. This was demonstrated by the Random Forest analysis which placed "business type" higher than "sign type", with the later grouping all private and corporate businesses into two large categories. Our more nuanced analysis of corporate businesses also showed how the subcategories influence language choice in disparate ways (national corporations favored the use of Spanish, while international counterparts favored the use of English). This highlights the importance of using smaller and more specific categories in the study of LLs, and also highlights the need to collect more data.

Our prediction of more English use in signs closer to the border was disconfirmed, providing further support for the importance of specific businesses in influencing language choice. In other words, visitors who come to CJ do so for specific services, regardless of the distance from the border. In addition, as the literature previously suggested (Baumgardner 2006, 2008; Gorter 2013; Rosenbaum et al. 1977; Hoffman 2017), the status of English as an international language, and its prestige, results in a prevalence of English across the LL of several locations, not only those closer to the border. It is therefore not only geographical proximity to the United States that makes English prevalent in the LL of CJ, but also the degree of superiority this language holds at the international level and the wish to attract American customers. This explains why the rate of English in the downtown area closer to the U.S. border is not significantly different to the rest of the city. While it is true that El Pasoans come to CJ to eat at restaurants and go to bars located in the downtown area, they will, when seeking medical services and beauty salons, drive the extra mile to find the best service, equal in quality to what is found in EP, at a lower cost.

Our third and last question investigated the implications of our findings for the vitality and status of Spanish and English in CJ. Our results showed a clear relationship between English use and SES, with the use of English increasing as a function of higher SES. This finding supports the idea that English has prestige in CJ, echoing previous studies about the status of English in Mexico (Baumgardner 2006, 2008). As stated earlier, there is a relationship between bilingualism and social class in Mexico, as Mexicans who can afford private education will send their children to bilingual private schools in CJ. However, being bilingual with a good proficiency in Spanish and English is something that only the most affluent classes in CJ can afford. Hence, social class and English proficiency are interrelated: belonging to a higher SES increases the chances of becoming bilingual, and becoming bilingual simultaneously increases the chances of climbing the social ladder.

There is a contradiction between the written presence of English and the oral use of this language in CJ. While the written presence of English in the LL of CJ is ubiquitous, the language is not actually spoken in the community (Teschner 1995). Spanish-English bilinguals do not use English when conducting their business in CJ, even when English is present in signs. This further supports the idea that English is not only used for functionality, but also for the prestige and value it assigns to a business. This finding runs counter to Landry and Bourhis (1997), who suggest that signs can signal which specific language is available for communication in a specific store, whether to receive a service or purchase a product. In the case of CJ, English is not used to signal that the service can be offered in English, but is instead motivated by marketing purposes and appreciated as a sign of prestige.

Despite the prestige of English in Mexico, the overall vitality of Spanish remains: while English is used for prestige and marketing purposes, Spanish has a strong symbolic power

that is associated with values of loyalty that construct Mexican identity (Hidalgo 1986). For instrumental reasons *juarenses* must learn English, but Spanish is part of their identity, and relinquishing it is not an option, meaning the maintenance of Spanish south of the border is safeguarded.

In sum, this paper has shown that analyzing the presence, visibility, and representation of different languages in a given territory can shed light on the power dynamics and social hierarchies within a society. CJ has proven to be the ideal place to conduct this LL study, both because it is a point of contact between Spanish and English, and because of its unique geographical position at the U.S.-Mexico border. The significant presence of English and Spanish-English bilingualism in CJ provided clear evidence of the superior status of English, which is used to assign prestige to products and services. Despite the ubiquity of English in CJ, the symbolic power of Spanish is indisputable. This symbolic power coupled with the large Spanish monolingual population in CJ; both play in tandem, guaranteeing the strong vitality of Spanish in the southern part of the U.S.-Mexico border, while keeping the prestige assigned to English intact.

**Author Contributions:** Conceptualization, N.M., N.M.O. and R.G.d.A.; methodology, N.M., N.M.O. and R.G.d.A.; validation, N.M., N.M.O. and R.G.d.A.; formal analysis, N.M.; investigation, N.M., N.M.O. and R.G.d.A.; resources, N.M., N.M.O. and R.G.d.A.; data curation, N.M.; writing—original draft preparation, N.M., N.M.O. and R.G.d.A.; writing—review and editing, N.M., N.M.O. and R.G.d.A.; visualization, N.M., N.M.O. and R.G.d.A.; project administration, N.M. All authors have read and agreed to the published version of the manuscript.

**Funding:** This research received no external funding.

**Institutional Review Board Statement:** Not applicable.

**Informed Consent Statement:** Not applicable.

**Data Availability Statement:** Data is unavailable due to privacy restrictions.

**Acknowledgments:** The authors wish to thank the two anonymous reviewers and the editors of the volume for their valuable feedback on earlier drafts of this paper. We are truly thankful to the students from LING/SPAN 3309 (Structure of Spanish) who helped us analyze some of the pictures in Spring 2023, to Joe Heyman for insights regarding census information and for sharing valuable data with us, and to Martin Lazzari for his support with the statistical analysis. All errors remain our own.

**Conflicts of Interest:** The authors declare no conflicts of interest.

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
