# Peer review of "Spanish Loyalty and English Prestige in the Linguistic Landscape of Ciudad Juárez, Mexico"

_languages, doi:10.3390/languages9020041_

Round 1
Reviewer 1 Report
Comments and Suggestions for Authors
This article presents an interesting and relevant issue we find in the borderlands with respect to the use of Spanish and English in the linguistic landscape. Other than some minor spelling revisions, is is ready to be published!

page 7 Table 1 first column: completes >> completos
page 9 Line 382 The United States >> the United States
page 17 Line 662 illustrate >> illustrate
Reviewer 2 Report
Comments and Suggestions for Authors
“Spanish Loyalty and English Prestige in the Linguistic Landscape of Ciudad Juárez, México” analyzes the LL in Ciudad Juárez the sister city of El Paso to understand the complex relationship between language use, prestige, and identity. The authors claim that the paper “investigates the social and linguistic factors that influence the use of English and bilingualism in public signage.” and that “therefore, the findings of this study contribute important insights for sociolinguistics, marketing, and economics”. I thank the authors for a well-written paper and the effort they put on showing the statistics that help us understand their results. However, I think that the paper needs some work before publishing.
First, I don’t think the title reflects what the paper is about. The authors mention the monolingual situation of CJ on several occasions (p. 1, 3, 20); if that is true, then, the Spanish almost-monolingual community of CJ does not have any other language to show its loyalty. However, there is not a recent demographics cited on the paper except one published on 1995.
Also, the authors mentions that English has prestige, which seems pretty obvious as it is an international, global language (a hyper-central language, according to Calvet 2006), but the use of English in the LL of CJ seems more due to the type of business and not just because of its prestige.
There are also some methodological problems with the data. The authors mentioned that pictures were taken in each of the wellness levels’ areas; however, how these pictures are equivalent to each other? Where were these taken? How many blocks were included? Were these taken in commercial, industrial, residential areas? Or data were collected in a geographical center of each area? No information is provided.
Furthermore, one of the main hypotheses of the paper is that areas closer to the border will show more use of English; however, “closer” is never defined in the text. Since closer, farther, are relative categories, the closeness to the border needs to be clarified. in other words, how close is closer?
The authors mentions that they only take into consideration English, Spanish and English/Spanish signals, but in the “Discussion and Conclusions session” they state, “we also coded and analyzed other languages present in signs around the city, but their proportion was so low (1.5%), that we excluded these from further analysis.” What were these languages? Were they Mexican/ Central American indigenous languages? If that is the case, it should be important to be mentioned.
Also, in the conclusions the authors “argue that English is used to […] attract border crossers from the U.S., who visit CJ for specific services.” How is this argument supported?? Is there any survey showing people crossing to CJ for specific business? For this reason, current studies in LL recommend conducting surveys/questionnaires/ethnographic studies among passers-by to know better how and why they modify and are modified by their LL, and to supplement the researcher’s perspective, which is always incomplete.
Finally, I would suggest authors to be careful when using initials. I could not find what the meaning of SES is as it is not explained the first time it is used.
